# Leveraging Wearable Sensors in Virtual Reality Driving Simulators: A Review of Techniques and Applications

**DOI:** 10.3390/s24134417

**Published:** 2024-07-08

**Authors:** Răzvan Gabriel Boboc, Eugen Valentin Butilă, Silviu Butnariu

**Affiliations:** Department of Automotive and Transport Engineering, Transilvania University of Brasov, RO-500036 Brasov, Romania; butila@unitbv.ro (E.V.B.); butnariu@unitbv.ro (S.B.)

**Keywords:** wearable sensors, driving simulator, virtual reality

## Abstract

Virtual reality (VR) driving simulators are very promising tools for driver assessment since they provide a controlled and adaptable setting for behavior analysis. At the same time, wearable sensor technology provides a well-suited and valuable approach to evaluating the behavior of drivers and their physiological or psychological state. This review paper investigates the potential of wearable sensors in VR driving simulators. Methods: A literature search was performed on four databases (Scopus, Web of Science, Science Direct, and IEEE Xplore) using appropriate search terms to retrieve scientific articles from a period of eleven years, from 2013 to 2023. Results: After removing duplicates and irrelevant papers, 44 studies were selected for analysis. Some important aspects were extracted and presented: the number of publications per year, countries of publication, the source of publications, study aims, characteristics of the participants, and types of wearable sensors. Moreover, an analysis and discussion of different aspects are provided. To improve car simulators that use virtual reality technologies and boost the effectiveness of particular driver training programs, data from the studies included in this systematic review and those scheduled for the upcoming years may be of interest.

## 1. Introduction

The development of devices used to monitor and manage people’s health has been the subject of intense medical study in recent decades [1,2,3]. Small, wearable devices with features that are nearly identical to those of stationary equipment have supplanted large, complex equipment. It is sufficient to recall the blood glucose analysis procedure. Rather than requiring the patient to be present for a blood test, wearable technology now allows for real-time blood glucose monitoring and the delivery of precise insulin dosages to the patient’s body. At the same time, the average life expectancy worldwide has significantly grown due to recent advancements in preventative and health insurance systems [4], meaning that a sizable portion of the elderly population is still active. Utilizing automobiles while in the driver’s seat is another aspect of this activity.

To prevent any harmful scenarios, it is evident that there is an abnormally high number of active drivers. Additionally, there is the issue of ensuring that drivers are not fatigued or in a condition of poor health when operating a vehicle.

Manufacturers of automobiles are attempting to equip their latest models with driver status identification systems. Naturally, testing these data collection systems on automobile simulators is a necessary first step. The virtual reality-powered automobile simulators need to be as dependable as possible to provide the most lifelike driving experience [5]. So, the participants can be fully immersed in a virtual environment that is highly comparable to the real one, with test findings on humans closely matching those from the actual test apparatus.

In recent times, wearable sensors have drawn a lot of attention due to their potential to improve virtual reality (VR) driving simulators [6]. Heart rate, galvanic skin response (GSR), electromyography (EMG), and eye tracking are just a few of the physiological signs that these sensors can monitor. These signals can reveal important details about the driver’s emotional state, stress level, and level of participation in the simulation.

Scholars have investigated diverse methodologies for incorporating wearable sensors into virtual reality driving simulations. These methods can be broadly divided into three groups:Sensor fusion: This method integrates information from several sensors to give a more thorough picture of the driver’s health. For instance, a driver’s degree of alertness can be determined by combining heart rate and GSR, and muscle tension can be identified using EMG.Real-time adaptation: This method makes use of sensor data to dynamically modify the simulation experience to better suit the requirements and preferences of each driver. For instance, simulation could lessen the difficulty of the driving task or offer extra help to a driver who appears stressed.Using sensor data, predictive modeling is an approach that attempts to forecast future driving behavior. For instance, potential diversions or dangerous driving behaviors may be predicted by examining heart rate and GSR patterns.

The integration of wearable sensors into VR driving simulations holds great promise for advancing the understanding of driver behavior and improving driving safety [7,8]. By combining the immersive capabilities of VR with the physiological and behavioral insights provided by wearable sensors, more effective driver training systems can be developed, identifying factors that contribute to driver error and developing predictive models to prevent accidents [9].

This paper provides a comprehensive review of the techniques and applications of wearable sensors in VR driving simulations. While the scientific literature contains a vast amount of research dedicated to reviewing existing work across diverse fields, like human emotion recognition [10], activity monitoring and motion control [11], biosensor technology in education [12], monitoring electrocardiograms (ECGs) [13], glove-based sensors [14] and so on, to our knowledge, there is no article summarizing the works focused on wearable sensors in the context of driving. 

Given the controlled environment that driving simulators provide [15], facilitating the effective use of wearable sensors, we opted to focus our search exclusively on studies employing this technology. Thus, the article aims to bridge this gap in the literature by exploring the potential of wearable sensors in VR driving simulations. We delve into existing techniques for integrating these sensors within VR environments and analyze their applications in driver assessment. By examining the current research landscape, we aim to shed light on promising avenues for future research and development in this dynamic field.

The following research questions were considered: What are the different types of wearable sensors that can be used in VR driving simulators?What are the different techniques for using wearable sensors in VR driving simulators?What are the different applications of wearable sensors in VR driving simulators?What are the challenges and limitations of using wearable sensors in VR driving simulators?What are the future directions of research in the use of wearable sensors in VR driving simulators?

### 1.1. Monitoring the Driver’s Activity and Health

Driving safely can be impacted by a variety of factors as it is an intricate undertaking that demands focus and coordination. It is crucial to continuously monitor the physical and behavioral health of drivers [16] since doing so can help lower the likelihood of accidents by spotting medical issues, indicators of exhaustion, and even incorrect driving conduct brought on by drug or alcohol use.

There is a wide range of equipment available today that can follow the behavior of drivers thanks to driver monitoring technologies [17]. From the perspective of the continuous health monitoring process, it is not just for drivers, but it empowers everyone to track health, prevent disease, and make healthy choices [18,19]. Also, continuous health monitoring is advantageous not just for people but also for the health system as a whole as it addresses issues about improved chronic illness management, improved medical problem prevention, and improved medical research.

Moreover, given the inherent dangers of driving, from the unpredictable actions of other drivers to pedestrian safety, the degree of ability to perform the activity under ideal circumstances must be continuously monitored. Certainly, there are also drawbacks of using wearable technology for continuous health monitoring. These include the expensive cost of the devices and related services, privacy concerns over data, and the possibility of measurement errors [20]. However, despite the challenges, the advantages of employing these techniques are compelling.

There are several kinds of sensors available for assisting driving:Safety-enhancing sensors: The detection of abrupt motions or alterations in posture that could indicate an accident is about to happen (inertial sensors). Braking systems or traffic partner warnings can be initiated based on these data.Driver assistance: using video sensors to track the driver’s eye movements to confirm that they are actively operating the car.Health monitoring: ECG, EMG, and sensors for indicators of stress, inattention, and drowsiness.Proximity sensors in vehicles: these can identify close objects and avert crashes.

Even with the potential uses of wearable sensors in the automotive industry, several obstacles still need to be overcome:Accuracy and dependability of sensor readings: driver monitoring and ADAS operations depend on accurate and dependable sensor readings.Data security and privacy: it is critical to maintain compliance with privacy laws and safeguard private behavioral and physiological data from unwanted access.Integration with car systems: for a consistent and easy-to-use user experience, wearable sensors must be seamlessly integrated with car systems, such as infotainment, dashboards, and ADAS.Driver acceptance and comfort: the widespread adoption of wearable sensors depends on their design being both driver-acceptable and comfortable to wear for extended periods.Standardization and compatibility: by creating industry guidelines for wearable sensors and how they integrate with car systems, we can encourage compatibility and interoperability amongst various brands and models.

More cutting-edge uses in the automobile sector could be anticipated since wearable sensor technology continues to be researched and developed, which will enhance comfort, safety, and the overall driving experience.

### 1.2. Operating Virtual Reality Driving Simulations

Driving simulations in VR have become a viable tool for driver education, research, and training. Comparing these simulations to conventional training techniques, there are many benefits, such as the following:Enhanced immersion: drivers can practice in a range of difficult situations without the hazards involved in driving in real life thanks to VR simulations, which offer a more realistic and immersive experience.Safety and cost-effectiveness: by lowering the chance of accidents and enabling greater repetition and practice, virtual reality simulations are both safer and more economical than on-road training.Adaptability and customization: driver requirements and preferences may be catered to in VR simulations, which offer individualized training and feedback.

Applications for VR driving simulators are numerous and include education and training for drivers [21], studies on driver drowsiness and distraction [22], disability rehabilitation [23], advanced driving assistance system (ADAS) development [24], and studies concerning driverless vehicles [25].

VR driving simulations are a quickly developing discipline, and research is always being conducted to further the realism, efficacy, and accessibility of these simulations. Prospective paths comprise immersions using several senses, customized fit and guidance, integration with field training, and use in law enforcement and public safety.

### 1.3. Comparing Virtual and Actual Equipment for Sensor Testing

There are two distinct ways to assess sensor performance: virtual testing and testing on actual equipment, each with pros and cons of its own. Sensor performance can only be accurately assessed by testing on actual equipment. For this testing, sensors are paired with real hardware, replicating their actual operating conditions and enabling a highly realistic performance assessment. Real equipment testing has the following benefits: it is accurate, yields findings that are realistic, and makes performance issues easier to identify. Testing on actual equipment has drawbacks such as high cost, difficulty and time commitment, and potential for sensor damage.

Evaluating sensor performance virtually is quicker and less expensive. Computer simulations of the sensors are used in this kind of testing. While simulation can imitate real-world scenarios where sensors would be employed, it is not as precise as testing on actual hardware.

### 1.4. Virtual Testing Benefits Include Speed, Low Cost, and Ease of Use

Virtual testing has several drawbacks, including lower accuracy compared to testing on actual equipment, difficulty in simulating specific real-world conditions, and the inability to pinpoint performance problems brought on by sensor interaction with the surroundings.

The following criteria are taken into consideration during the sensor testing process to choose the best test method: accuracy, prices, time, and the requirement to replicate unique test conditions. On the other hand, real testing is advised if great accuracy is required, and virtual testing is the goal for minimal expenses.

## 2. Materials and Methods

In this section, we describe the methodology used to carry out the literature review. A review article typically employs a defined and transparent protocol to identify articles relevant to a specific area of research by conducting a thorough examination of the scientific literature. This work followed the established statement and quality standards of PRISMA 2020 [26]. The search process was conducted by one author, but the selection of papers to be included in the analysis was performed by consensus between all of the authors. 

### 2.1. Search Methodology and Scope

The objective of the review was to identify scientific papers presenting studies that use wearable sensors for driver monitoring in virtual environment settings. Therefore, the first established goal was to find out what can be found in the specialized literature on this topic. For this, the research questions presented in the Introduction were defined. After that, research was carried out to identify the key terms to be applied in the search strategy. The keywords and their combinations about wearable sensors for driving in virtual reality were the following: wearable AND sensor* AND (driver OR driving OR vehicle) AND (virtual OR “virtual reality” OR simulator). The keywords were applied to the title, abstract, and keywords fields. The search was conducted on 4 January 2024.

TITLE-ABS-KEY (human AND sensor* AND (driver OR driving OR vehicle) AND (virtual OR “virtual reality” OR simulator)) AND (LIMIT-TO (LANGUAGE, “English”)) AND (LIMIT-TO (DOCTYPE, “ar”)) AND (LIMIT-TO (PUBYEAR, 2013) OR LIMIT-TO (PUBYEAR, 2014) OR LIMIT-TO (PUBYEAR, 2015) OR LIMIT-TO (PUBYEAR, 2016) OR LIMIT-TO (PUBYEAR, 2017) OR LIMIT-TO (PUBYEAR, 2018) OR LIMIT-TO (PUBYEAR, 2019) OR LIMIT-TO (PUBYEAR, 2020) OR LIMIT-TO (PUBYEAR, 2021) OR LIMIT-TO (PUBYEAR, 2022) OR LIMIT-TO (PUBYEAR, 2023)).

### 2.2. Eligibility Criteria 

The following eligibility criteria were applied to the included articles: (a) published in a peer-reviewed journal; (b) published in the last eleven years (from 1 January 2013 to 31 December 2023); (c) written in English language; (d) reporting the use of wearable sensors; (e) containing user study experiment in virtual environment; (f) addressing automotive field, more specifically driving; (g) available in full text. 

### 2.3. Information Sources 

The search was performed in 4 scientific databases, using different queries adapted according to the input requirement of each of them. The selected databases were Scopus, Web of Science, Science Direct, and IEEE Xplore. All of these databases index the most relevant journal and conference articles for various fields of life sciences, social sciences, physical sciences, and health sciences.

### 2.4. Data Extraction 

Articles that met the inclusion criteria were reviewed by two authors [RGB, EVB]. The articles were imported into EndNote software (EndNote 20.6, Thomson Reuters) and the full papers were analyzed to extract the relevant information. Beyond data related to the identification of the study (title, authors, year), data extracted include information from the following summary: source of publication, country, number of articles per year and number of citations, final number of participants in the sample and their characteristics (sex; age mean, range, and standard deviation; social background), objective of the study, measures of outcomes, future directions, limitations/biases stated and remarks for interesting information that the study provide.

### 2.5. Summarizing and Reporting 

Data extraction was performed in tabular form, using Microsoft Excel 365. The essential characteristics of the selected papers were extracted, and the main findings were summarized and discussed. When there were differences in interpretation, they were resolved through discussions between all three authors. In addition, a quality assessment of the studies was performed to ensure that the results would not be significantly altered. For this, the Critical Appraisal Skills Programme (CASP) Qualitative Studies Checklist [27] was used, which provides a standardized instrument composed of ten key questions to assess the scientific relevance of research articles. The evaluation was performed by two authors and the results are presented in Figure 1 (the numbers of the studies on the x-axis correspond to the numbers of the references from Table 1). As can be seen, all the selected articles were included in the review since they had a low risk of bias.

## 3. Results

After searching all the above keywords and applying the filters mentioned above according to the defined criteria, 295 articles were obtained, which consisted of 35 papers from the Scopus database, 76 papers from the Web of Science database, 145 papers from the Science Direct database, and 39 papers from the IEEE Xplore database. A number of 38 of these papers were duplicates and were consequently removed. The remaining 257 articles were screened by title and abstract (65% of them by RGB and 35% of them by EVB) to verify that they met the proposed eligibility conditions. The inconsistencies were resolved via discussion with the third author [SB]. After this step, 213 sources were excluded, and the remaining 82 papers were screened via full-text reading. This stage was also performed by the first two authors and discrepancies were discussed with the third author. Finally, 44 articles were considered suitable to be included in the review. Figure 2 shows the process of searching and selecting data sources.

The literature search resulting in a list of 44 articles revealed that they cover a range of 9 years (2015–2023). In the first two years of this period, only one study was published. Then followed an increase in the number of published articles, and most studies were published in 2020 (n = 9). The number of citations also varies by each year, with a peak in 2020 as well. The next peak is in 2017, but this is due to work [28], which is the most cited article among all 44 (according to Scopus, on the date of publication of this work). Figure 3 shows the trend regarding the number of publications and the number of citations per year. 

According to the data collected from the 44 studies included in the review, they correspond to a geographically wide range of 14 different countries. However, a concentration of the research is evident in in the USA and China (each of them contributing eight papers), followed by Germany (six papers), Japan (five papers), and the UK, the Republic of Korea, and Australia (three papers each). The remaining countries have only one study, as can be shown in Figure 4. This distribution points towards a potential gap in research participation from other parts of the world, and future efforts could aim to broaden the geographical reach of this field’s investigation.

Based on the source of publication, the journals *Transportation Research Part F: Psychology* and *Behaviour and Accident Analysis and Prevention* ranked first with eight published papers each, followed by *Sensors*, with five articles, and *Transportation Research Part C, Expert Systems With Applications* and *IEEE Transactions on Intelligent Transportation Systems* (with two papers each). The other journals published a single study on the selected topic. The great majority of journals are from the Elsevier publishing company (61.36%), followed by MDPI (13.63%) and IEEE (13.63%). The results are presented in Figure 5.

The total number of participants involved in the included studies was 1506. The age of the participants ranged between 18 and 70+ years old. The age range of participants was not specified in 18 studies. Also, in nine studies, the mean age and the standard deviation (SD) were not mentioned. For this reason, it was not possible to calculate a mean age for the whole sample considering the incomplete reporting within the studies. Figure 6 presents the mean age and SD of the participants from the selected studied. Please note that the numbers of the studies on the x-axis correspond to those in Table 1.

Also, the gender of the participants was not reported in all the studies. Six studies did not include this information. Of the participants who were involved in the studies that reported this information, 972 (67.97%) were males, 457 (31.96%) were females, and only one study reported that one of the participants was non-binary. The distribution of the participants according to gender is presented in Figure 7. 

Through the systematic literature search, 44 papers closely related to sensor-based wearable systems for monitoring human drivers were found. In particular, there were eight papers related to driver drowsiness, eight papers regarding driver stress, four papers that evaluated comfort/discomfort, four papers that assessed driver workload, and others related to takeover performance (n = three), motion sickness, fatigue, driving performance (n = two), visual distraction, inattention, boredom, and so on. An overview of the parameters that were analyzed in the selected works and the sensors used for this purpose is presented in Figure 8 and Table 1. The abbreviations shown in the legend of the figure are used further in the article. 

Concerning the topic of wearable system technology, different physiological signals were explored. EDA and PPG were used in 15 studies each, followed by ECG (13 papers), ET (9 papers), EEG (7 studies), MT (7 studies), EMG (2 studies), and EOG (2 studies). 

Although they are very similar in meaning, people perceive and interpret differently the words “sleepiness,” “fatigue,” “tiredness,” and “drowsiness” and they cannot be used interchangeably [29]. Drowsiness at the wheel often stems from a lack of sleep [30]. For drowsiness detection, the HRV signals were used in [31]. They were acquired using wearable ECG and PPG sensors. In [32], the drowsiness detection was performed using a machine learning approach applied to physiological data collected from a non-intrusive wrist-worn sensor, achieving an accuracy of over 92%. In the same way, the feasibility of using wrist-worn wearable devices for accurate driver drowsiness detection was demonstrated in [33]. Another approach based on anomaly detection was proposed in [34]. The HRV features were extracted from RR interval (RRI) data using an ECG sensor attached to the body of the user. The fluctuation in the R-R interval using a self-attention autoencoder was also exploited in [35]. The data were collected using an ECG, and the sleep conditions were validated by specialists based on an EEG. The NN intervals were used in [36] to estimate the HRV. In [37], a watch-type activity monitor sensor was used to detect the driver’s drowsiness by monitoring the activity data. Finally, both motional and physiological information was measured in [38] using a self-developed wearable device. 

Sleepiness, as a less advanced state than drowsiness, which conveys to the body the need to sleep soon, was investigated in [39] using an objective sleepiness scale based on an EEG and EOG. Sleepiness was considered in the paper as the opposite of vigilance. Fatigue is another factor that impairs driving. It is similar to sleepiness, but it may include muscle weakness or decreased motivation. The main cause of fatigue could be monotonous driving, sleep deprivation, or boredom [40]. Fatigue was monitored in [38] using a wearable device for measuring the driver’s physiological and behavioral information, including PPG, GSR, temperature, acceleration, and rate of rotation. In [40], driver fatigue was predicted using an ECG and breathing waves. Also, boredom was investigated in [41] and a gamified boredom intervention was proposed to reduce unsafe behaviors. The boredom state was recorded using EDA and ECG signals. 

Regarding stress, it has become a major concern in modern society due to its significant impact on various aspects of well-being, including driving safety [42]. Various wearable biosensors were used in the literature to measure stress. In [43], an EEG headband was designed to investigate the effect of autonomous driving algorithms on brain activity. The authors found that the estimated stress is higher in manual driving compared to autonomous driving. A framework based on EEG patterns for the identification of driving-induced stress was proposed in [44]. Another authors have used a combination of physiological signals to detect driving stress: ECG, GSR, and respiration [28], EDA, PPG, and BVP [45], and PP and EMG [46]. 

Driving is a complex cognitive task that demands various mental functions, and driving performance is in relation to task demands [47]. Especially in monotonous and very complex situations, the mental workload should have high values [48]. Also, it was shown that mental demands impair situation awareness [49]. Therefore, it is important to predict the driving workload to ensure safety and to increase driving performance [50]. An approach based on machine learning for workload prediction was presented in [51]. The physiological data were recorded by low-cost sensors using a PPG-based method. In [50], the prediction was performed using eye-tracking metrics. To classify the driver’s mental workload, the physiological signals were gathered using ECG and EDA signals in [52].

The wearable sensors were also used to study human comfort in autonomous vehicles [53], to evaluate the impact of a driving condition prompt system on passenger comfort [54], to identify physiological indicators of discomfort in automated driving [55], and discomfort caused by prolonged driving [56]. Also, driving performance in manual driving [57,58], or takeover performance in conditionally automated driving [59,60,61] were performed using different methods based on wearable devices. 

Regarding these wearables, the most used device in the selected papers was the Empatica E4 wristband [62] (used in eight studies), a device that incorporates EDA and PPG sensors. Tobbi Pro Glasses [63] were used in four papers for eye tracking, and Shimmer3 GSR+ [64] was used in three papers for GSR and PPG measurements.

**Table 1 sensors-24-04417-t001:** Characteristics of the selected studies.

No.	Study	Assessed Parameter	Measuring Method	Device
1	Affanni, A. et al. [43]	Stress	EEG	Self-developed EEG headband
2	Akiduki, T. et al. [34]	Inattention, body movement, drowsiness	ECG, MT	ATR_Promotions compact wireless sensor TSND121
3	Beggiato, M. et al. [55]	Discomfort	EDA, ET	Handset control, smartband Microsoft Band 2, SMI Eye Tracking Glasses 2
4	Bitkina, O. V. et al. [50]	Workload	ET	Tobii Pro Glasses 2
5	Chen, H.-Y. W. et al. [65]	Training for conditionally automated driving	ET	Tobii Pro Glasses 2
6	Chen, K.-T. et al. [59]	Takeover performance	ET	Tobii Pro Glasses
7	Chen, L.-I. et al. [28]	Stress	ECG, EDA (GSR)	EKG electrodes, elastic Hall effect sensor
8	Choi, M. et al. [38]	Stress, fatigue and drowsiness	PPG, EDA (GSR), MT	Self-developed
9	Du, N. et al. [60]	Takeover performance	EDA (GSR), PPG	Shimmer3 GSR+ unit
10	Du, N. et al. [61]	Takeover performance	EDA (GSR), PPG	Shimmer3 GSR+ unit
11	Fujiwara, K. et al. [35]	Drowsiness	ECG, EEG, EOG	Grapevine Neural Interface Processor system
12	Giot, C. et al. [39]	Sleepiness	EEG, EOG	International 10/20 electrode system—EEG, portable digital recorder (Dream^®^, Medatec)
13	González-Ortega, D. et al. [58]	Driving performance	ECG, EMG, EDA (GSR), MT	Shimmer electrocardiogram sensor
14	Guo, Y. et al. [54]	Comfort and motion sickness	MT	Smartphone equipped with an inertial sensor and an accelerometer
15	Halim, Z., Mahma, R. [44]	Stress	EEG	EEG non-invasive cap EMOTIV EPOC +
16	Heikoop, D. D. et al. [49]	Mental demands	ECG, ET	AD Instruments PowerLab26T—ECG, Dikablis Professional head-mounted eye tracker
17	Karjanto, J. et al. [66]	Motion sickness	PPG-HR	Pulse sensor
18	Kosuge, R. et al. [67]	Driving performance	MT	Objet electronic device from ATR-Sensetech
19	Kraft, A.-K. et al. [68]	Driver behavior	EDA	skin conductivity, electrodermal activity—EDA, four-camera eye-tracking system (SmartEye 6.1)
20	Kundinger, T. et al. [32]	Drowsiness	PPG-HRV	Empatica E4 wristband
21	Kundinger, T. et al. [33]	Drowsiness	PPG-HRV	Smartwatches and Empatica E4 wristband
22	Lee, H. et al. [31]	Drowsiness	ECG, PPG	Two body-worn sensors: a Polar H7 strap ECG sensor and an MS Band2 PPG sensor
23	Lee, J. et al. [69]	Calm breathing	EDA, ECG-BR	Zephyr BioHarness chest strap and the Affectiva Q-sensor wristband
24	Li, X. O. et al. [70]	Human posture	sEMG	DELSYS testing system
25	Ma, Y. et al. [71]	Driving risk	PPG, EDA	Empatica E4 wristband
26	Paschalidis, E. et al. [72]	Stress	EDA-SCR, PPG-HR	Empatica E4 wristband
27	Paschalidis, E. et al. [45]	Stress	EDA-SCR, PPG-HR, BVP	Empatica E4 wristband
28	Peruzzini, M. et al. [47]	Workload	ECG-HR; ET	Zephyr BioHarness 3.0, Tobii Pro Glasses 2
29	Qian, K. et al. [37]	Drowsiness	MT	A portable environmental sensor (2JCIE-B), a watch-type activity monitor (actigraph; ambulatory monitoring)
30	Scherz, W. D. et al. [73]	Stress	ECG-HR	Chest band Polar H10 and the wristband Empatica E4
31	Schneider, L. et al. [56]	Motion activity, muscle stiffness, discomfort	MT	Full-body human motion capture system by Xsens Technologies B.V.
32	Schwarz, C. et al. [36]	Drowsiness	PPG-HR, BVP	Empatica E4 wristband
33	Seet, M. et al. [74]	Attentional control	EEG	64-channel waveguard™ EEG cap (ANT Neuro, the Netherlands)
34	Steinberger, F. et al. [41]	Boredom	EDA, ECG	Biopac BioNomadix3 MP150WSW system was used with BN-PPGED and BN-ECG2
35	Su, H., Jia, Y. [53]	Comfort	EEG, EDA, PPG-BVP; SKT	Empatica E4 wristband and the Emotiv EPOC+ headsets
36	van Gent, P. et al. [51]	Workload	PPG, EDA -GSR	Low-cost sensors powered by an Atmel ATMega328p embedded processor board
37	Wang, S. et al. [75]	Visual distraction	ET	Pupil Labs eye tracker
38	Wei, W. et al. [52]	Workload	ECG, EDA	BIOPAC MP160
39	Young, K. L. et al. [57]	Driving performance	ET	Google glass
40	Zhang, X. et al. [76]	Collision prediction	EEG	64-electrode cap Neuroscan
41	Zheng, R. C. et al. [46]	Stress	PP, EMG	Active electrode—EMG signals, digital perspiration meter
42	Zhou, F. et al. [40]	Fatigue	ECG-HR, BR; ET	Bio-Harness 3.0 sensor, ISCAN eye-tracking goggles
43	Zhu. A. et al. [77]	Brake response time	TV	AR63A/AS63A vibrometer
44	Zou, X. et al. [78]	VR evaluation	SA	HMD Oculus Rift

## 4. Discussion

Numerous sensors can track a driver’s physical and even behavioral conditions in a simulator. Other significant issues that can be identified include the fusion of data measured by the sensors, data interpretation, and the computer’s decision-making process, in addition to the issues that arise when mounting these sensors on a human body and impacting the comfort of the driver.

The combination of data from several sensors and the real-time processing of those data suggests the need for improved hardware and software systems, which drives up costs. Simultaneously, new software with high performance must be developed.

There are clear decision-making procedures when it comes to other types of sensors that monitor other health parameters or the behavior of the driver. For instance, if a sensor monitoring the functioning of the heart detects information necessitating the immediate stopping of the vehicle, the decision-making process is relatively simple. In these situations, it is imperative to follow defined protocols that are based on the combination of input from several sensor types and the development of a series of incremental judgments that the car’s computer can implement. To ensure that all potential scenarios are covered, these procedures must be created by interdisciplinary teams that include medical professionals.

The age range of the subjects is another observation about the evolution of simulator tests. Figure 6 illustrates this, showing that the subjects’ average age is almost exactly thirty years old. This represents a small issue for the researchers, who appear to have depended on participants in the form of students or colleagues for their experiments. In the actual world, there are persons who drive cars who are beyond 70 or 75 years old, with the average age of drivers being significantly higher. Medical statistics indicate that these individuals are among the most susceptible categories, making them more likely to suffer from specific conditions related to the safe driving of automobiles.

It is important to keep in mind though that the process of researching and developing new sensors is highly dynamic, and very soon, considerably more advanced and effective sensors may be available for purchase. This may result in the development of VR-based automobile simulators that are far more lifelike and akin to real cars, giving drivers a greater sense of comfort and confidence.

To test under typical driving situations, these simulators require the creation of many scenarios including the entire spectrum of stimuli, including scenarios with impending danger. It is important to outfit these simulators with top-notch 3D visualization and motion perception apparatuses.

There are no references in the specialized literature that we looked at about how test settings and constructive elements affect the data that the sensors gather. It is hard to say if the subject’s discomfort can provide different datasets.

Based on the analysis of the existing research results, this paper addresses five research questions. The findings are summarized in the following sections.

### 4.1. RQ1. What Are the Different Types of Wearable Sensors That Can Be Used in VR Driving Simulators?

The following sensor technologies for measuring driver activity were identified in the analyzed studies (they are presented in descending order, depending on the number of articles in which they are used):EDA (electrodermal activity) or GSR (galvanic skin response): measures skin conductance;PPG (photoplethysmography): measures changes in blood volume;ECG (electrocardiogram): measures heart activity;ET (eye tracking): measures eye movements;EEG (electroencephalography): measures electrical activity in the brain;MT (motion tracking): measures body movements;EMG (electromyography): measures muscle activity;EOG (electrooculography): measures electrical changes associated with eye movements.

### 4.2. RQ2. What Are the Different Techniques for Using Wearable Sensors in VR Driving Simulators?

Following the analysis, we could identify four types of techniques used in the studies for analysis with wearable sensors:Single-sensor analysis: analyzes data from a single sensor to assess a specific aspect of driver behavior.Multi-sensor fusion: combines data from multiple sensors to provide a more comprehensive understanding of driver behavior.Real-time analysis: analyzes data and provides feedback to the driver in real time.Offline analysis: analyzes data after the simulation has ended.

### 4.3. RQ3. What Are the Different Applications of Wearable Sensors in VR Driving Simulators?

Regarding the paper’s aim, the following categories could be summarized:Driver monitoring—identify and assess driver states such as the following:
○Stress—a feeling of mental and physical strain caused by a demanding situation [79]; driver stress, characterized by faulty decision making, manifests in aggressive behaviors like sudden accelerations and decelerations [80], significantly increasing the risk of accidents [81]; ○Drowsiness—a more intense state of sleepiness, characterized by a strong urge to fall asleep [82]; it is considered a phase of transition between wakefulness and sleep [83];○Comfort/discomfort—often considered two endpoints of a single dimension [53], such that the comfort is the lack of discomfort [54]; both are complex concepts affected by physical, physiological, and psychological factors (e.g., motion sickness, sound, climate, feeling of safety, etc.) [84,85]; ○Mental workload—the cognitive demand of a task [86] or the amount of processing capacity that is used for task performance [47]; so, driving performance is closely related to the mental workload of the driver [52]; mental workload can have a visual component and it is named visual workload in this case [75], and it can be evaluated using psychophysiological components, task performance, and self-rating questionnaires [87];○Driving performance—the observable actions and/or behaviors of the driver while carrying out the designated tasks behind the wheel; the parameters used in the research to assess the driving performance include lateral control, longitudinal control, reaction time, gap acceptance, eye movement, workload measures [57,58,88], and takeover performance for conditionally automated driving [59,60]; ○Motion sickness—physiological and/or psychological discomfort arising from the perceived or actual passive movement of the human body [89]; it is mainly a subjective phenomenon [90] and it is usually evaluated using subjective scales [66]; ○Fatigue—a feeling of tiredness or exhaustion [91] that can be caused by sleep deprivation (sleep-related) or by the demands of driving itself (task-related) [92]; it can be detected based on driver behavior or physiological measures [38]; the fatigue and drowsiness are interdependent [93], but there are distinctions between their definitions [94];○Visual distraction—anything that takes the driver’s visual field away from the road ahead [95]; it can be assessed using eye-tracking technology [75];○Inattention—occurs when the driver’s focus strays from the road for some non-compelling reasons [96]; the detection of inattention can be made using vehicle signals or the driver’s physiological and behavioral signals [34];○Boredom—an unpleasant feeling of wanting to do something stimulating [97]; it is caused by the general characteristic of a driving task, which is generally monotonous or predictable [41]; ○Human posture—the position and alignment of the body while seated behind the wheel; it can be measured using different data from wearable sensors such as acceleration sensors [98], physiological, pressure sensors, and multi-sensor fusion [70,99].Factor investigation—investigate the effects of different factors on driver behavior: the impact of peripheral visual information in alleviating motion sickness [66], the influence of training paradigms on improving driver knowledge [65], or the influence of sonification feedback on takeover events [59] in conditionally automated driving, the effects of mental demands on situation awareness [49], the impact of an inaccurate self-assessment on driving behavior [67], the effect of stress on gap-acceptance decisions [72], the effects of system failures on the behavior of drivers [68], the influence of virtual reality headsets for driver stress [73], the effects of a seat-integrated mobilization system on motion activity [56], the influence of boredom and gamification on driver behavior [41], the impact of Google Glass on driving performance [57], or the effects of presence, arousal, and task workload on user experience [78].Safety optimization—develop safer driving strategies and interventions: eye-tracking technologies to predict and classify the perceived driving workload [50], predict driving risk on curved roadways [71], and investigate how visual distraction influences driving safety [75].

### 4.4. RQ4. What Are the Challenges and Limitations of Using Wearable Sensors in VR Driving Simulators?

Across the 44 studies, a range of challenges and limitations were identified, as follows:Sensor-related issues and challenges—sensors and cables can be uncomfortable and constrain natural movements, impacting driving behavior [28,32,37,47,56]; the sensitivity of the devices to various factors, such as wearable physiological sensors can be affected by body movement, lighting conditions, or other environmental factors [53,69,73];Data collection challenges—limited sensor range that cannot capture all relevant information [68]; some factors can influence data recording that are not present in simulated environments, like the vibrations in real vehicles [32]; the difficulty in separating signal from noise in physiological data [53]; the difficulty to implement complex system architectures in real driving scenarios [37]; the complexity of the set-up, that can be burdensome and time consuming [47]; and the rigor in placing and calibrating the sensors [74];Generalizability—the lack of generalizability due to small datasets [37]; the limited realism of VR simulators compared to real cases [40,74,78], but it should be noted that simulated experiments are complementary research approaches to the real-world driving [77]; and the need for combining more physiological and contextual data [53];Measurement limitations—difficulties in inferring some parameters (e.g., mental state, time pressure) from physiological data [72]; the limitation of a single measurement method [52]; and some measured variables could have different physiological responses, so a full range of experiences should be considered [40].

Besides these challenges, we should also mention the cost of implementation because wearable devices as well as the acquisition and the processing system are not very accessible. At the same time, the collection and use of driver data raise some privacy concerns.

### 4.5. RQ5. What Are the Future Directions of Research in the Use of Wearable Sensors in VR Driving Simulators?

To meet the difficulties and challenges presented above, and also to expand the research, the authors also proposed various future research directions, which can be summarized as follows:The development of more advanced and accurate sensorial systems—developing comfortable, user-friendly wearable sensors with high-quality data transmission [28,32]; integrating data from various sensors (EEG, ECG, EDA, etc.) to have a more comprehensive picture of the driver state [43,53]; exploring the use of additional wearable sensors [66]; including other stimuli to induce different emotions in the driver [44,69]; comparing commercial fitness devices and medical-grade equipment [32]; investigating the influence of body movement on the quality of measured signals [33]; inducing different levels of a certain measure (e.g., workload, boredom) [41,69]; and using more accurate systems of data collection [36];Improving data analysis methods and algorithms—refining models with larger datasets and real-world testing [33,35]; developing methods to quantify different levels of a measure from physiological signals [44]; enhancing algorithms for the real-time processing of sensor data [53]; and increasing signal quality and data transmission [28];Enhancing VR simulations—testing the effectiveness of VR simulation and sensor-based assessments in real-world driving scenarios [38,60]; developing VR simulators that better replicate real-world experiences [32]; incorporating environmental data in the simulations [53]; investigating novel methods that combine VR simulations with real-world driving experiences [65]; assessing driving tasks also in automated driving scenarios [33]; and investigating the potential of VR itself to induce stress and its impact on research results [73];Study design and generalizability—conducting studies with larger samples of participants with different ages, driving experience levels, and so on [33,34,61,73,78]; collecting additional self-reported measures to better interpret sensorial data [60]; testing the proposed methods in various driving situations [65]; applying the developed approaches in different applications or fields beyond driving [35,73,78]; combining data from multiple simulator studies [36]; considering more metrics to be assessed related to driver states [65]; and studying the influence of various environmental factors on driver behavior [78].

### 4.6. Wearable Sensors Limitations

Although wearable sensors for driving are still in their infancy, they have the potential to enhance comfort, safety, and the driving experience in general. Currently, automobiles are equipped with a number of sensors that serve different purposes:Enhancing physiological sensors to monitor blood pressure, perspiration level, and heart rate in order to identify signs of exhaustion, stress, or drowsiness and head and eye tracking to gauge alertness and identify driving-related distractions.Customization and enhancement: biometric verification, comfort modifications, and enhanced performance.

The installation and operation of these sensors provide new difficulties with regard to wearable device data privacy, integration, and standardization for various automobile models and a number of technological constraints, including sensor accuracy and battery life and reliable data transmission.

Currently, a number of automakers are testing wearable technology-based or in-car systems to address vehicle fleet management (commercial vehicle driver behavior and fatigue monitoring) and biometric identification (using smartphones as car keys through fingerprint or facial recognition) fields.

In order to provide a more customized, safe, and comfortable driving experience, wearable sensors and in-car technologies are probably going to work together in the future of wearable sensor systems. Although wearable sensors are getting more and more advanced, there are still significant differences in their precision, accuracy, and latency that can be caused by a number of different variables. For instance, modern heart rate monitors are often reliable (±5%) during rest and during moderate- to high-intensity exercise, but accuracy may drop at these intensities.

The positioning of the sensors is another crucial consideration. Improper positioning, such as fitting the sensors more loosely on the body, can seriously affect the accuracy of sensors like activity trackers and heart rate monitors. Certain accelerometer-type sensors may overestimate or underestimate data if a subject moves their body repeatedly. This is why certain sensors need to be calibrated for the best results. This could imply the need for systems that are more intricate and the requirement for more manual or semi-automated processes, which would make driving less comfortable.

Wearable sensors usually have low latency when it comes to recording and data transfer speed; most data are often transmitted in milliseconds. Nevertheless, there may be a small delay when processing and sending data to a smartphone app.

There are two processes involved in the calibration process for wearable sensors, albeit they can differ based on the brand and kind of sensor:Factory calibration: Pre-calibration is a feature of most wearable sensors. For most users, this basic calibration should be adequate.User calibration: for increased accuracy, several sensors (such as accelerometers, magnetometers, and GPS units) allow users to carry out their own calibration.

For the best results, the user may need to repeat particular motions or positions multiple times during the calibration process. Depending on how frequently the action is performed, how long it takes, and whether or not specialized calibration instruments are needed, this could be upsetting to a driver.

One important component affecting user comfort is the utilization of wearable sensors. In addition to the knowledge that these sensors can enhance driving performance, control, or health monitoring, any factor that causes discomfort of any kind can cause an emotion to arise throughout the wearable sensor design process:Physical design: material (pliable, breathable materials like soft fabrics or medical-grade silicone); weight and size (lighter and smaller sensors are generally more comfortable); pressure points on the body (evenly distributed weight and a comfortable fit); and form factor (considering the human body’s biomechanics).Skin contact: non-allergenic materials are used; breathable materials are used; ventilation features are improved to remove perspiration; and features are simple to clean.Aesthetics and durability.

Wearable sensor designers can better realize the potential of wearable technology and improve user acceptance by prioritizing these criteria when designing devices that people can wear comfortably for extended periods of time. Also, there are some issues related to simulation scenarios. For instance, some wearable sensors could be sensitive to environmental factors [53]. However, some researchers argue that physiological sensors are less vulnerable to environmental conditions [38,40] . Also, it is important to combine more information in order fully understand the status of the driver because the environmental factors could affect even the driver’s stress or workload [47,72]. The environmental conditions are often omitted in the studies. In the current analysis, there are only five papers considering environmental factors, such as weather conditions [44,52,59,65], or temperature, humidity, illuminance, and noise level [37].

### 4.7. Autonomous Vehicles 

One application that can be made to the present generation of automobiles is the usage of wearable sensors. On the other hand, autonomous vehicles represent another area of research—and perhaps a path toward development—that is evident on a global scale. This will mean that the driver of a classic vehicle will have less and less accountability for their health and engagement while they drive their vehicle. Among the 44 selected articles, 17 explored autonomous driving scenarios with different levels of automation implemented.

The use of autonomous vehicles is expected to bring about several benefits, including the following:Enhanced safety: Since human error is the primary cause of traffic accidents, autonomous cars can remove it. They are thought to be able to cut accidents by up to 90%.Enhanced efficiency: There would be less congestion and longer travel times, and there would be more fluid traffic. Fuel consumption and emissions could be decreased by autonomous vehicles by synchronizing speed and optimizing routes.Accessibility: individuals without the ability to drive due to age or medical conditions would be able to move around independently.

But, there are also a lot of difficulties:Technological development: in order to ensure safety and dependability, self-driving technology must undergo extensive testing.Infrastructure: Autonomous vehicles cannot yet operate on the current infrastructure. Road signs, traffic signals, and traffic lights would all need to be modified.Regulations: Autonomous vehicles are not covered by the present legal system. Laws specifically addressing traffic and liability in the event of an accident would be required.Population acceptance: The general population is reluctant to embrace driverless vehicles. A campaign of information and education is required to boost trust in this new technology.Employment impact: professional drivers may lose their jobs as a result of autonomous vehicles.Ethical concerns: the development of autonomous vehicles brings up ethical concerns, like the problem of hacking and moral quandaries in emergency situations.Costs: At the moment, autonomous cars are highly costly. For the general public to have access to them, the expenses must be lowered.

The following estimations paint a picture of these level 4 and 5 vehicles’ probable market share: by 2030, 90–130 million autonomous vehicles should be in use [100].

### 4.8. Views, Perspectives

Road safety, comfort, and the driving experience could all be completely transformed by wearable driver sensors. We should anticipate seeing even more creative uses of these sensors in the automobile sector as technology advances.

Here are a few particular uses for wearable driver sensors:Insurance firms offer premium savings to drivers who use wearable sensors to track their health.The incorporation of wearable sensors by automakers to enhance safety and offer driver support.Creating more efficient traffic safety regulations by utilizing wearable sensors to gather information on driver behavior.

## 5. Conclusions and Limitations 

In this article, we summarize the state-of-the-art literature on sensors attached to drivers in VR settings to monitor their state or behavior. The paper provides insights into wearable sensors and virtual driving simulators. It explores the specific sensor types used, data collection methods, applications, and the challenges and future research directions identified in experiment-based studies.

To ensure transparency and minimize bias in our selection process, we conducted this systematic review following the PRISMA guidelines. We identified relevant articles through rigorous searches in relevant scientific databases, applying specific search criteria. This approach resulted in a total of 44 studies. It is important to acknowledge the inherent limitations of this review. The first limitation is publication bias. Studies with non-significant findings are less likely to be published in high-impact journals, potentially leading to their exclusion. Additionally, the limited number of studies meeting the defined specific criteria can restrict the generalizability of our conclusions. Moreover, relevant information might exist in non-indexed literature, although it may have methodological or quality limitations.

In addition, our search covered a defined timeframe from 1 January 2013 to 31 December 2023. This ensures a focused analysis of recent research, but the studies published outside this range are excluded, so possible relevant articles are missing.

Despite these limitations, this systematic review provides valuable and insightful findings that contribute to the current understanding of the topic of wearable technology applied in the field of driving simulations.

## Figures and Tables

**Figure 1 sensors-24-04417-f001:**
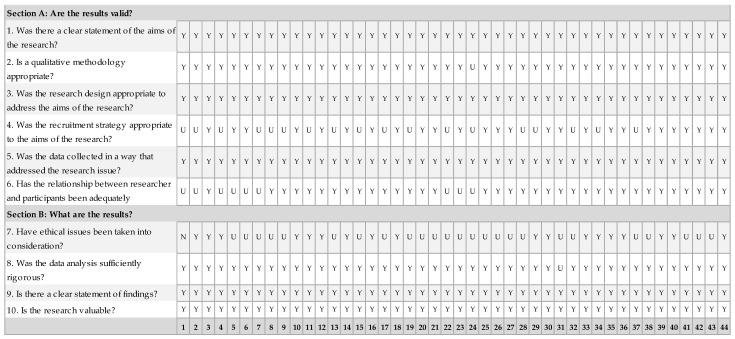
Quality assessment of the selected studies using CASP quality study checklist. Note: Y—yes, N—no, U—unclear.

**Figure 2 sensors-24-04417-f002:**
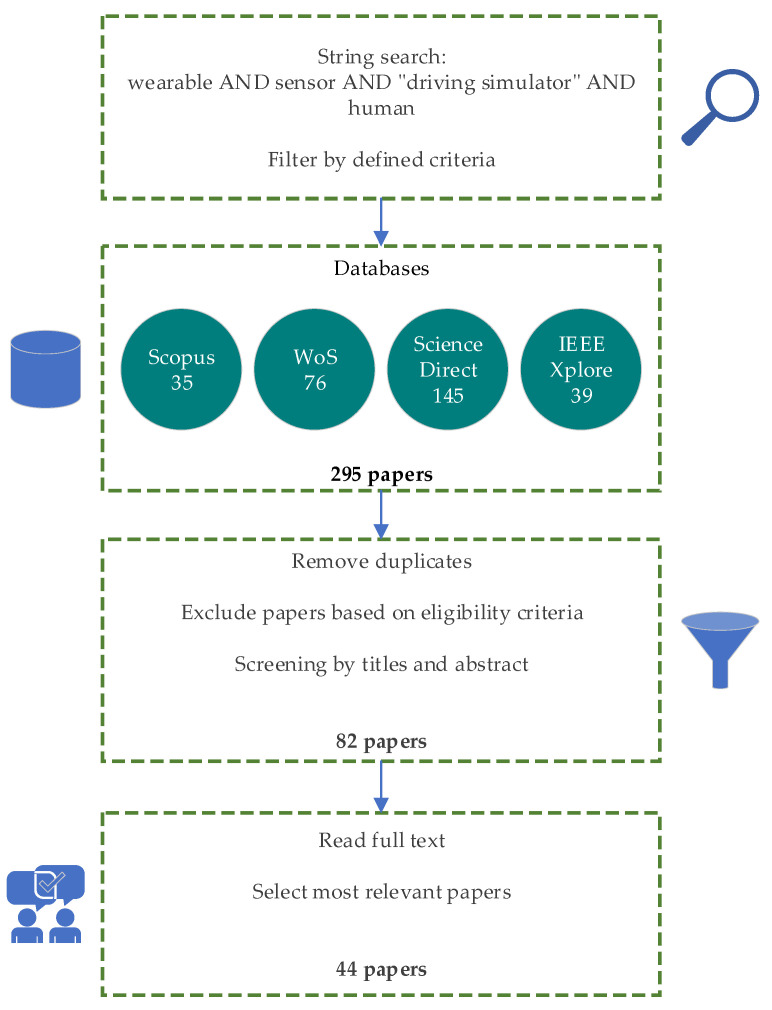
Flowchart of the literature screening process.

**Figure 3 sensors-24-04417-f003:**
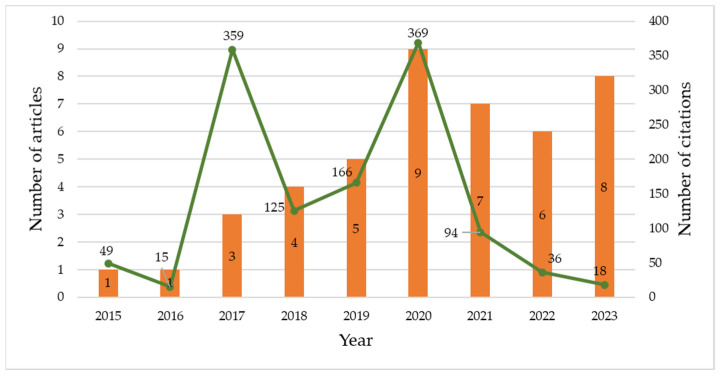
Evolution of annual publication and number of citations per year.

**Figure 4 sensors-24-04417-f004:**
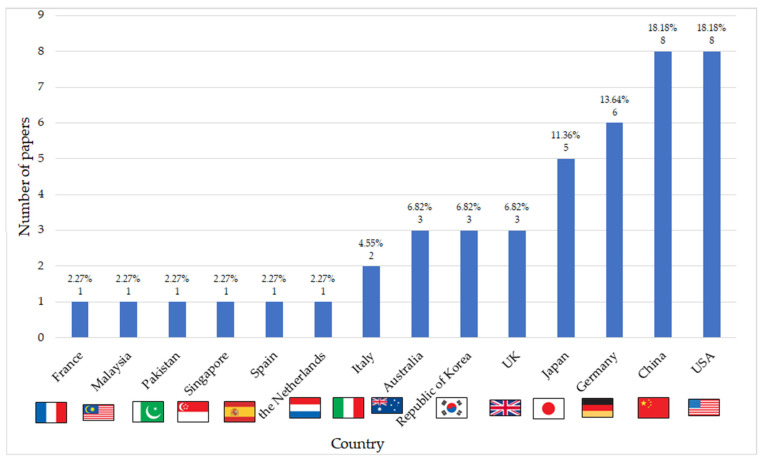
Geographic distribution of publications.

**Figure 5 sensors-24-04417-f005:**
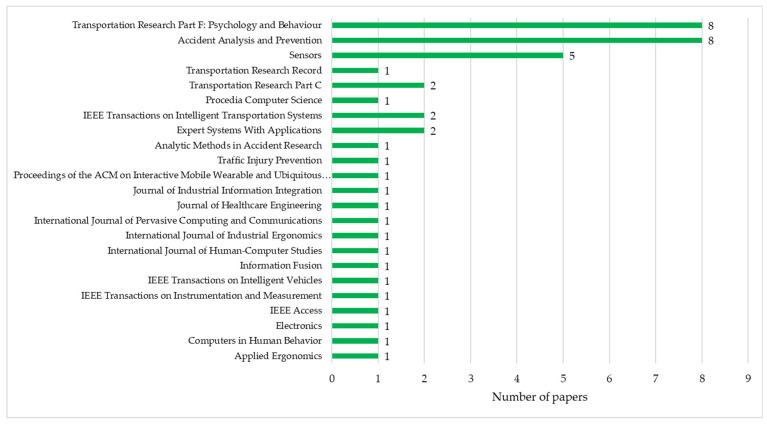
Number of papers based on the source of publication.

**Figure 6 sensors-24-04417-f006:**
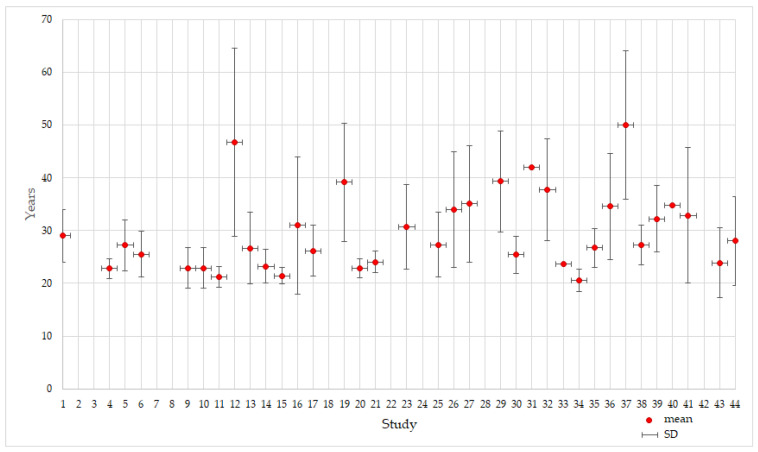
Mean age and standard deviation of the participants in the selected studies.

**Figure 7 sensors-24-04417-f007:**
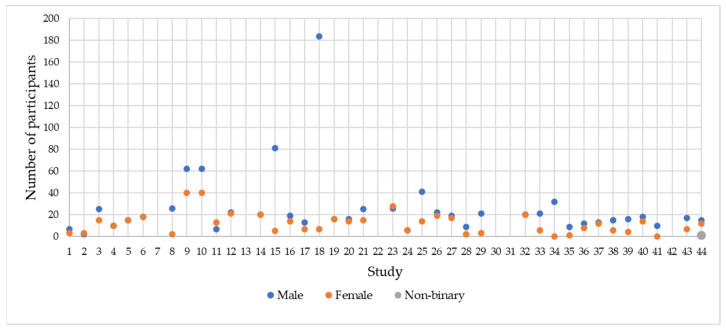
The gender of the participants in the selected studies.

**Figure 8 sensors-24-04417-f008:**
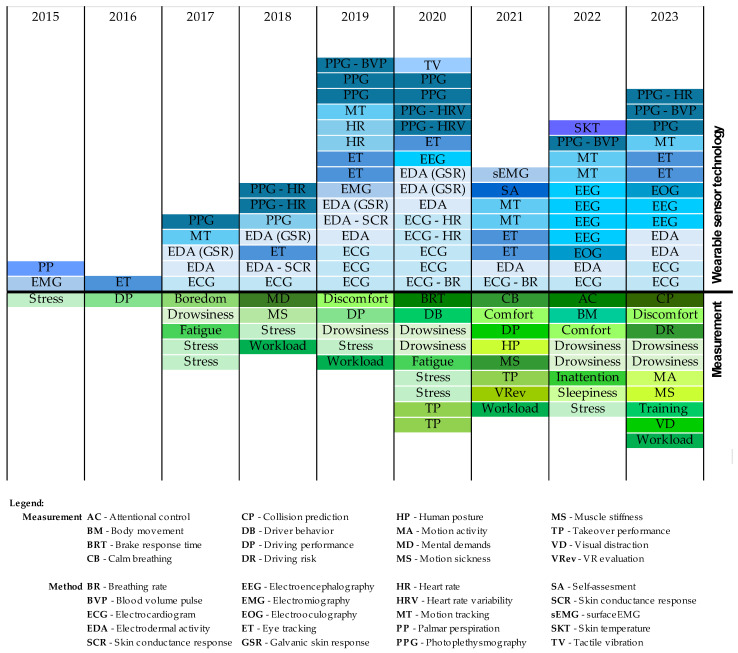
Wearable devices and measurements in the selected studies.

## Data Availability

The dataset is available on request from the authors.

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
