# Peer review of "Leveraging Wearable Sensors in Virtual Reality Driving Simulators: A Review of Techniques and Applications"

_sensors, 2024, doi:10.3390/s24134417_

Round 1
Reviewer 1 Report
Comments and Suggestions for Authors
In this review paper, the authors selected a significant number of publications (44 studies, between 2013 and 2023) to review the techniques and applications of leveraging Wearable Sensors in Virtual Reality Driving Simulators.
The paper is consistent, well documented and provides a comprehensive review of the techniques and applications of wearable sensors in VR driving simulations.
Regarding the limitations of wearable sensors, the paper should give more details about the following aspects:
- technical limitations - accuracy, precision and latency;
- user experience: calibration procedure (if necessary);
- issues related to user comfort, the influence of sensors on the driver;
I also believe that the issues related to the simulation scenarios should be investigated, considering external Interference, environmental conditions and simulation setup.
Comments on the Quality of English LanguageMinor editing of English language required.
Author Response
Comment 1: The paper is consistent, well documented and provides a comprehensive review of the techniques and applications of wearable sensors in VR driving simulations.
Response 1: Thank you very much for your appreciation. We will carefully consider your feedback to refine the paper.
Comment 2: Regarding the limitations of wearable sensors, the paper should give more details about the following aspects:
- technical limitations - accuracy, precision and latency;
- user experience: calibration procedure (if necessary);
- issues related to user comfort, the influence of sensors on the driver.
Response 2: Thank you for your suggestion. A new subsection related to the mentioned aspect was added in the Discussion section (4.6. Wearable sensors limitations).
Comment 3: I also believe that the issues related to the simulation scenarios should be investigated, considering external Interference, environmental conditions and simulation setup.
Response 3: Thank you for your suggestion. A new paragraph was included in the end of the Discussion section (Lines 632-640).
Reviewer 2 Report
Comments and Suggestions for Authors
The overall content of the paper is out of proportion to the number of references, so it is suggested to continue to add the main text to describe the current research progress in detail, which is the core of the review.
However, in general, the manuscript is relatively detailed, and there are some expressions that can be further optimized and maintain the unity of the whole article.
Comments on the Quality of English LanguageOverall can
Author Response
Comment 1: The overall content of the paper is out of proportion to the number of references, so it is suggested to continue to add the main text to describe the current research progress in detail, which is the core of the review.
Response 1: Thank you for your suggestion. We added a new section in the Discussion section (4.6 Wearable sensors limitations).
Comment 2: However, in general, the manuscript is relatively detailed, and there are some expressions that can be further optimized and maintain the unity of the whole article.
Response 2: Thank you for your observation. The manuscript has been thoroughly reviewed and phrasing has been improved for clarity and unity.
Reviewer 3 Report
Comments and Suggestions for Authors
This study addressed an important gap in the literature by examining the potential of wearable sensors in VR driving simulators using A literature review performed on databases as Scopus, Web of Science, Science Direct, and IEEE Xplore. This article was well written, and the study was well designed. I only have a few minor comments to further improve the manuscript.
1. I suggested included the item Information sources as 2.3 and mention about the main multidisciplinary and engineer-focused databases which were contemplated. Databases as (example): ASCE Library, INSPEC, Safety Lit, Sage Journals, Science Direct, Scopus, Springer, Taylor & Francis, TRID - Transportation Research Board, Web of Science, and Wiley Online Library….
2. 2.3 Suggestion: Data items
I suggested mention that beyond data related to the identification of the study (title, authors, year), data extracted will include information from the following summary: leading institution and country, description of the driving simulator- sensors, type of the road scenario (number of lanes, directions), number of scenarios, route length, number of drives per participant, ordering of scenarios/studied elements, presence or absence of a practice trial before the principal test, trial duration, break duration, final number of participants in the sample and their characteristics (sex; age mean, range, and standard deviation; social background), sampling method, requirements and justification for the sample, if participants were excluded and the motives and methods of exclusion, objective of the study, measures of outcomes, analyzed road geometry feature, main methods and results, limitations/biases stated and remarks for interesting information that the study might provide...
NOTE: It is not necessary all of them, but it is necessary present more characteristics on text.
3. The authors did not provide any analysis
4. Figure 8 and Table 1 are are presented in good condition and very clearity.
Other than my above comments, I believe the manuscript is in great condition.
Author Response
Comment 1: I suggested included the item Information sources as 2.3 and mention about the main multidisciplinary and engineer-focused databases which were contemplated. Databases as (example): ASCE Library, INSPEC, Safety Lit, Sage Journals, Science Direct, Scopus, Springer, Taylor & Francis, TRID - Transportation Research Board, Web of Science, and Wiley Online Library...
Response 1: Thank you for your suggestion. An item called Information sources was included describing the selected databases.
Comment 2: 2.3 Suggestion: Data items
I suggested mention that beyond data related to the identification of the study (title, authors, year), data extracted will include information from the following summary: leading institution and country, description of the driving simulator- sensors, type of the road scenario (number of lanes, directions), number of scenarios, route length, number of drives per participant, ordering of scenarios/studied elements, presence or absence of a practice trial before the principal test, trial duration, break duration, final number of participants in the sample and their characteristics (sex; age mean, range, and standard deviation; social background), sampling method, requirements and justification for the sample, if participants were excluded and the motives and methods of exclusion, objective of the study, measures of outcomes, analyzed road geometry feature, main methods and results, limitations/biases stated and remarks for interesting information that the study might provide...
NOTE: It is not necessary all of them, but it is necessary present more characteristics on text..
Response 2: Thank you for your observation. A detailed description of the extracted data was provided as you suggested.
Reviewer point #3: The authors did not provide any analysis.
Response 3: Thank you for pointing out this issue. A new section was included in the Discussion chapter.
Comment 4: Figure 8 and Table 1 are are presented in good condition and very clearity.
Response 4: Thank you for your remark. We appreciate your feedback to improve the paper.
Comment 5: Other than my above comments, I believe the manuscript is in great condition.
Response 5: Thank you for your valuable comments and for your help to refine the manuscript.